# A systematic analysis of regression models for protein engineering

Richard Michael[1][☯], Jacob Kæstel-Hansen[2][☯], Peter Mørch Groth[1,3], Simon Bartels[1], Jesper Salomon[3], Pengfei Tian[3], Nikos S. Hatzakis[2], Wouter Boomsma[1]*

**1** Department of Computer Science, University of Copenhagen, Copenhagen, Denmark, **2** Department of Chemistry, University of Copenhagen, Copenhagen, Denmark, **3** Enzyme Research, Novozymes A/S, Kongens Lyngby, Denmark

☯ These authors contributed equally to this work.

* wb@di.ku.dk

**Data Availability Statement:** All code required to replicate the results can be found at https://github.com/MachineLearningLifeScience/protein_regression. All data, including computed

## Abstract

To optimize proteins for particular traits holds great promise for industrial and pharmaceutical purposes. Machine Learning is increasingly applied in this field to *predict* properties of proteins, thereby guiding the experimental optimization process. A natural question is: How much progress are we making with such predictions, and how important is the choice of regressor and representation? In this paper, we demonstrate that different assessment criteria for regressor performance can lead to dramatically different conclusions, depending on the choice of metric, and how one defines generalization. We highlight the fundamental issues of sample bias in typical regression scenarios and how this can lead to misleading conclusions about regressor performance. Finally, we make the case for the importance of calibrated uncertainty in this domain.

## Author summary

Supervised machine learning is increasingly used to predict the function and properties of proteins. The performance obtained with these methods relies on a multitude of factors including how data is represented, how observations are distributed, how training is conducted, and how performance is measured. In this paper, we systematically assess the importance of these different components in a protein regression pipeline. We discuss the benefits of using representations extracted from protein language models, the impact of the choice of regression algorithm, and the role of uncertainty. Finally, to avoid misleading performance claims, we stress the need for carefully aligning the train/test setup to reflect the setting in which the prediction algorithm will ultimately be applied.

## Introduction

Accurate prediction of protein traits related to function and stability remains an important challenge, both for *in silico* protein engineering and for assessing the phenotypic consequences of genetic diseases [1–3]. Recent years have seen progress in *unsupervised* models for

representations are archived and publicly available at https://erda.ku.dk/archives/9a379e8618a1ba1f2730ec33fa3a736d/published-archive.html.

**Funding:** This work was in part supported by the Danish Data Science Academy (to RM ddsa.dk, DDSA-PhD-2022-010 which is funded by the Novo Nordisk Foundation, NNF21SA0069429, novonordiskfonden.dk, and VILLUM FONDEN, 40516, veluxfoundations.dk). Further funding includes the NNF Center for 4D cellular dynamics (to NSH, NNF22OC0075851, novonordiskfonden.dk) and Villum Synergy (to NSH and WB, veluxfoundations.dk, DeepDesign 40578), the Innovation Fund Denmark (to WB and PMG, innovationsfonden.dk, 1044-00158A), the MLLS Center (Basic Machine Learning Research in Life Science, novonordiskfonden.dk, NNF20OC0062606), Digital Pilot Hub (to SB, Skylab Digital, Danish Ministry of Education and Science), and the Pioneer Centre for AI (to RM, PMG, SB, WB, Danish National Research Foundation, dg.dk, grant number P1). The funders played no role in study design, data collection, analysis, decision to publish, or preparation of the manuscript.

**Competing interests:** The authors have declared that no competing interests exist.

predicting the fitness effect arising from protein variants. For single protein families, statistical models of aligned sequences have been used successfully to characterize the effect of mutations through a change in (approximate) likelihood [4–10] and attention-based language models have since extended this idea beyond single protein families [11–14].

While unsupervised models are very useful descriptors of variants with respect to a natural baseline, protein engineering often requires us to go beyond the unsupervised setting. One reason is that the function optimized for will often differ from the biological function selected for by evolution [15]. Another is that we generally wish to improve our predictive capabilities as we make more experimental observations of a system, which requires that we train models in a *supervised* fashion.

Supervised learning is associated with a number of challenges. First of all, it requires us to carefully consider how the available data are applied, since they are now used both for training and evaluation. The data can be split into training and test sets in different ways, and the similarities we permit between these sets will influence our assessment. The goal should be to choose a test set that reflects the data distribution that we expect to see when the regressor is ultimately applied in practice. But this matter is complicated by the fact that in an optimization setting, regressors are frequently used partially out-of-domain, i.e. to make predictions on sequences further from the wildtype than those in the training set, or distributed differently. A representative test set might not be available in this case. Even the notion of *in-domain* and *out-of-domain* is often not clear cut, and can depend on the choice of how a protein is encoded. The embeddings computed by protein language models (PLMs) are for instance likely to have a biologically richer notion of similarity than a distance based merely on amino acid identity. The extrapolative setting also highlights the requirement for regression algorithms to provide reliable uncertainty estimates. This is important to ensure that downstream decisions, i.e. by human practitioners of optimization algorithms, can be based on an informed trade-off between the predicted fitness of proposed candidates and the uncertainty of these estimates. Finally, supervised learning on protein sequences is made difficult by the fact that data sets for a particular protein system are often very limited in size, and can have substantial sample biases.

In this paper, we systematically describe the challenges involved in supervised learning for protein engineering. We describe the constituents of a protein regression pipeline (Fig 1),

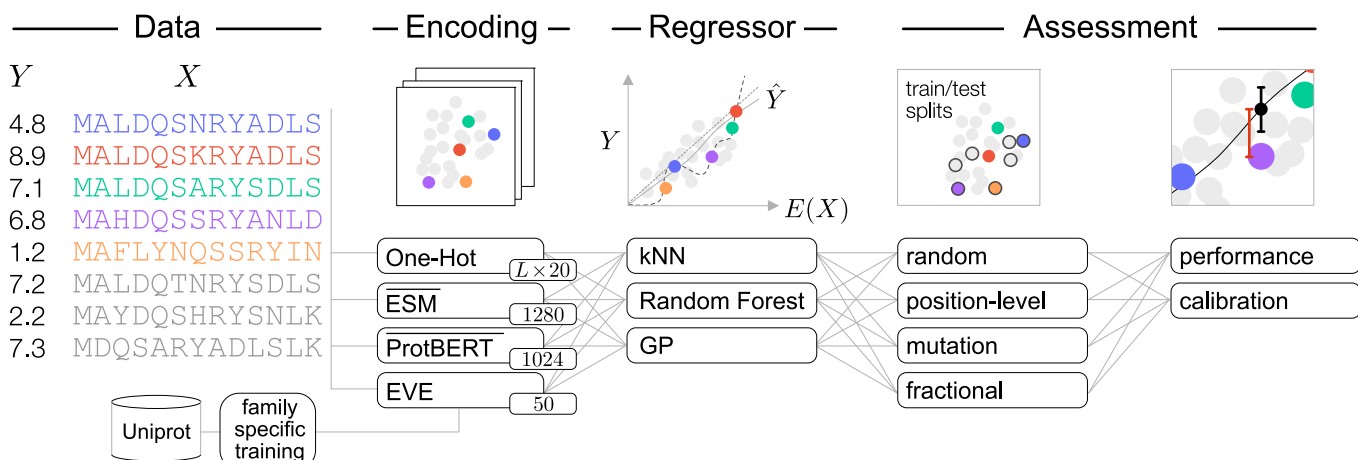

**Fig 1. A general overview for supervised learning tasks on protein systems.** For each protein exists labelled experimental assay *Data*, multiple sequence alignments (MSA), and sequence representations (a subset of the *Encoding*s annotated with the respective dimensionality). The language model (i.e. ESM-1B and PROTBERT) representations are the positional means of the internal representation (indicated by the bar). We fit and assess regressors, and select training and validation data through different protocols (*Assessment*). From the performance and calibration results we assess an in-silico optimization task for variant selection.

placing particular focus on the assessment stage. We show empirically that different choices at this stage can have a dramatic impact on the conclusions drawn, and discuss how we can design train/test splits to best reflect the generalization capabilities we desire for a given task. We also investigate the role of embeddings of the input protein sequences, comparing several recent language models with simpler alternatives. Finally, we consider the metrics used to measure performance, and highlight the importance of including calibration of predicted uncertainties when comparing regression procedures.

## Results

The fundamental objective in protein engineering is to find modifications, *variants*, of naturally occurring amino acid sequences that optimize one or multiple properties of interest, *e.g.* the thermostability of an enzyme [16, 17]. Since there are twenty naturally occurring amino acids, the space of possible amino acid sequences of a given length $L$ is $20^L$, an astronomically large number even for short proteins. In practice, engineering R&D pipelines often restrict themselves to sequences that deviate by only a few mutations from a naturally occurring, *wild-type*, sequence (WT). Exhaustive search through this restricted space is typically still intractable. In practice, we are limited in time and resources, and the search strategies employed are often based on combinations of random sampling and the intuition of domain experts. Developing better optimization strategies has therefore become an active area of research [18–21].

As part of such optimization strategies, the need arises to *predict* fitness values for unseen protein sequences. One example is the *surrogate function* in a Bayesian optimization setting, which predicts the expected fitness value and its uncertainty of any candidate protein. Together with an *acquisition function* which specifies a trade-off between exploration and exploitation, this allows us to select new candidates optimally, according to a well-defined protocol. Our success in optimizing proteins thus depends critically on the ability to regress the fitness of a protein against its sequence. Rather than working with the raw amino acid sequence, the input can be encoded in various ways to facilitate modelling. A simple choice is ONE-HOT-encoding, where each amino acid in a protein sequence or alignment is encoded in a $d$-dimensional vector, where $d$ is the number of different amino acid labels (typically 20 plus gap symbols and nonstandard labels). In recent years, it has become clear that regression performance can often be enhanced further by regressing against learned protein representations [13, 22–24]. This can potentially simplify the search space and make it possible to extrapolate further away from the wildtype sequence.

The next step is to choose a suitable regression algorithm. This choice will depend on factors such as 1) appropriate inductive bias, 2) expressivity relative to the amount of available data, 3) attributes of a model such as the ability to predict uncertainties. Ultimately, a dominant reason for the choice will be that a particular model is perceived to perform better on the specific task of interest. As we discuss the pros and cons of different regression algorithms and representations, it is thus of critical importance that we can reliably assess this performance. In contrast to unsupervised fitness prediction, assessment in the supervised setting will require us to split the data into train, validation, and test sets. Depending on the application task of interest, different choices are meaningful, and this choice can have a substantial impact on the reported performance. The performance will depend on the nature of the fitness landscape, and the amount of available data. Since experimental assays typically produce only tens up to a few thousands of measurements at a time, protein regression typically resides in the low-data regime, thus restricting the capacity of regression algorithms we can meaningfully apply. The experimental assays can also be of different quality, have different levels of coverage, and display more or less correlation to the functional trait of interest.

These are thus the key components in a protein regression pipeline: the nature of the data, its encoding, the regression algorithm, and the assessment strategy (Fig 1). We will now consider them in turn.

## Data

On the input side, protein fitness data can have different scope, ranging from sampling broadly from naturally occurring sequences, sometimes referred to as wildtype exploration, to sampling sequences locally around a single or a few naturally occurring sequences. Since our study focuses on protein optimization, we consider the latter category here, but refer to related work for considerations about wildtype exploration [24, 25].

Experimental techniques for characterizing protein properties, particularly many high-throughput assays, are often imperfect proxies of the desired functional trait and exhibit considerable noise. For instance, a common first step in a protein engineering pipeline is a site-saturation mutagenesis experiment, which in a single experiment characterizes the majority of variants deviating by a single mutation from the wildtype protein. Apart from relatively high levels of noise, these initial high-throughput experiments often have only limited correlation to the functional trait of interest, and are therefore sometimes augmented with lower-throughput, more costly assays in later stages of the optimization process. After an exploration of single mutants, protocols will often proceed with variants with greater separation from the wildtype. In these later stages, there will typically be considerable selection bias in the experimentally probed sequences, as a result of new variants being selected based on results from previous experiments. These effects have consequences for the performance we can expect from our regression algorithms, and in the design of our split strategies, as we will see below.

In addition to the sequences with observed experimental values (*i.e. labelled* sequences), there are typically many sequences available for similar proteins from the databases of naturally occurring proteins. If the functional trait we are optimizing for is related to the biological function of the protein, we can hope to gain additional information from these unlabelled sequences, for instance by learning improved representations or by unsupervised predictions.

To ground our discussion on modelling choices in regressor design, we conduct an empirical evaluation on a representative set of protein variant effect data sets. The data sets were selected to vary in data quantity, in how far the observed variants are from their closest wildtype, and to be diverse respective to their biological organismal functions and structures. The experimental data we consider are from mutational scans, which recorded growth under different stresses depending on the protein systems (see Table A in S1 File cf. [8, 14]). For each synthesizable candidate sequence we observe a measurement $y \in Y \subset \mathbb{R}$, as a label.

For our initial experiments, we will use two commonly employed train/validation split scenarios: a random cross validation (Random CV) strategy where the data is split at random (uniformly), and a position-level cross validation (Position CV), where splits are made by positions in the chain (non-overlapping segments are assigned to either the training, validation or test group). We will return to the choice of splitting procedures in our treatment of assessment techniques later in the paper.

## Protein representations

Advances in *representation learning* have shown that different approaches to unsupervised learning from biological sequences lead to differently structured representations [8, 12, 13, 22, 26, 27]. *A priori*, it is not clear how such differences in structure will translate into downstream performance. Therefore, the usefulness of a representation can only be assessed in terms of the performance of a downstream prediction algorithm. For the particular protein systems

investigated here, we show such an analysis in Fig 2A, comparing the ONE-HOT representation to two recent large language model representations, ESM-1B [11, 28] and PROTT5 [26, 29], and a family-specific representation extracted from the EVE variational autoencoder model [10]. We will return to the exact choice of metric and cross validation scheme later, and focus first on the fact that regression performance is clearly heavily impacted by the choice of representation (Fig 2A, bottom). As suggested by the striking results reported for fitness prediction with large language models and family-specific density models in the zero-shot (unsupervised) setting [14, 26], we might expect increased performance when using the corresponding representations as input for supervised training. Indeed, we observe that representations are capable of increasing regression performance beyond using the raw amino acid sequence. Although the strength of this effect depends on the protein, choice of regressor, and cross validation scheme, the representations extracted from language models seem to generally be a robust choice (Fig 2A). Interestingly, we note that this does not hold true for all choices of learned representations. In particular, we observe consistently that the latent space of a protein family-specific variational autoencoder (here EVE) seems to underperform in downstream regression tasks. This is somewhat surprising, since the *evo-score*, which is the evidence lower bound (*i.e.* a proxy for the likelihood against the *WT*) extracted from the same model seems to fare better on average, despite being a one-dimensional value. We find that the performance across PLM models is consistently high. We expected more recent, larger language models to outperform their earlier counterparts and indeed found a significant gain from PROTBERT to PROTT5. Surprisingly, this was not generally the case for ESM-1B, ESM-1V, and ESM-2 (see Section 3.2 in S1 File).

Some of the most striking differences in performance occur across the two different splitting strategies, an issue that we will revisit in detail later in the paper. The usefulness of the different representations also varies significantly between protein systems. This presumably reflects the different nature of the assays. For instance, the cases where the evo-score is predictive of the assay value are presumably those where the assay is a good proxy for the biological function.

One final remark on the choice of representation relates to their dimensionality. The high dimensional per-position embeddings produced by language models are not well suited for direct input to a regression model. Often, the representations are aggregated over the length of the protein by simple averaging. For simplicity, this is also the strategy employed here, although we stress that better choices exist, and can have considerable impact on performance [22]. Even when employing such an averaging strategy, one is left with a representation of (often more than) hundreds of dimensions. For certain regression algorithms, like Gaussian Processes using kernels based on Euclidean distance, such high dimensional spaces are known to be challenging. For regressors for which the number of parameters scales with input size (*e.g.* a simple linear regressor), large representations might also lead to concerns of overfitting. In an ablation study we considered whether an unsupervised dimensionality-reduction pre-processing of the averaged representations could have a positive effect on regression performance, but generally found this to be detrimental (see Fig W in S1 File). However, we note that several recent protein optimization studies have found it beneficial to include in their optimization protocol a supervised dimensionality reduction step trained specifically on the task [30, 31].

## Regression algorithms

Choosing the right regression algorithm is often considered as the most critical of the model design choices in the pipeline. System-specific protein regression typically operates within the *low-data regime*, which implies that particular care must be taken in the choice of model class

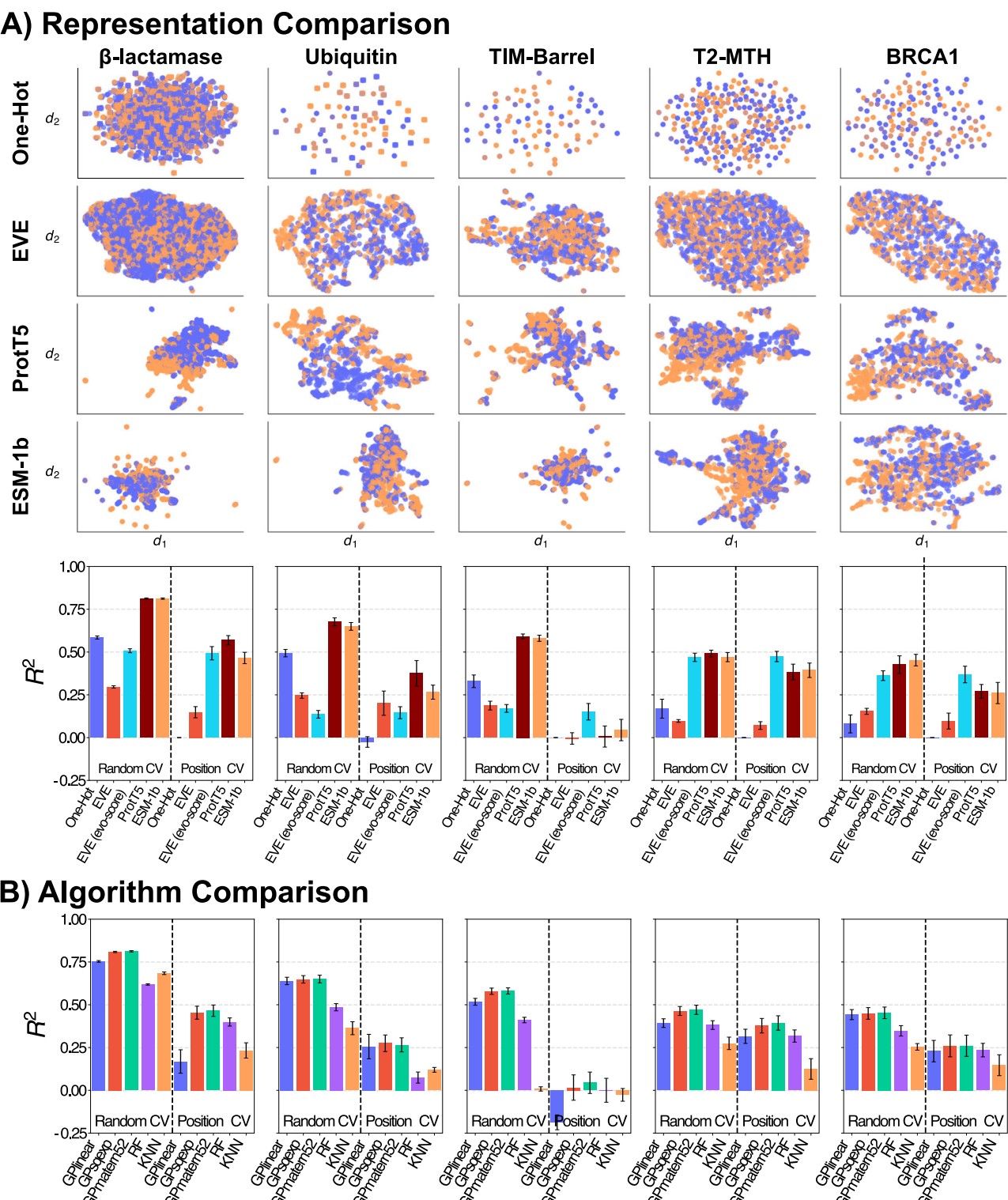

**Fig 2. Representation and regressor comparison.** We assess five distinct protein data sets by: A) Top: two-dimensional spatial separation of UMAP reduced representations (axes are UMAP output dimensions) color-coded in functional (orange) and non-functional (purple) variants. Bottom: performance of a GP-Matérn$\frac{5}{2}$ regressor on the representation as input reporting accuracy (adjusted $R^2$) on the test set of the CV protocol with bars as standard-error across splits. B) All supervised regressors on the ESM-1B representation input. Additional representations and results are in Figs A and B in S1 File.

                    

and model capacity to avoid overfitting to the noise in the data. In our case study, we select a set of representative methods, that are expected to show reasonable performance in this regime: (1) a $k$NN regressor, (2) a *Random Forest* (RF) with predictive ensemble uncertainties, and (3) three *Gaussian Process* (GP) regressors (linear kernel, squared exponential kernel, and Matérn$\frac{5}{2}$ kernel—see details (parameters and regressor optimization) in Methods). Each regressor is implemented to provide not only a predicted value $\hat{y}_i$, but also a predictive variance estimate as an associated uncertainty $\hat{\sigma}_i^2$ (see Methods).

When comparing the performance of these regressors across the five data sets we find the GP regressors perform well despite the high dimensional input when hyperparameters are optimized (Fig 2B). The fact that a GP with linear kernel in many cases provides a reasonable performance suggests that much of the signal in this limited data regime can be captured with a linear model, but this will depend on the system in question, the amount of available data, and the learned representation. A more general observation is that the choice of hyperparameters is at least as important as the choice of regressor; if chosen poorly, it can have a significant detrimental impact on performance. Note that we focused here on regression performance using a language model representation, but find similar effects with other choices of representations. For details on both points, see Fig V in S1 File.

As in the previous section, we again observe a dramatic decrease in performance when we change splitting strategy from random sampling to position-level splitting (*i.e.* predicting in sequence regions unseen during training). The size of this effect is system specific. In cases like the TIM-BARREL, we have no predictive ability in the position-level split, although the random split performance is on par with other systems. We will explore this dependency in detail below.

## Assessment: Metrics, tasks, domains, and generalization

**Metrics.** An elementary part for any learning algorithm assessment is a means to assess predictive performance. To measure performance comparatively across data sets and systems we use the adjusted ratio between residuals and total residuals: $R^2 = 1 - \frac{\frac{1}{N}\sum_i^N (y_i - \hat{y}_i)^2}{\frac{1}{N}\sum_i^N (y_i - \mu_y)^2}$, which indicates an improvement of our predictions $\hat{y}$ over the training-set mean as a baseline $\mu_y$ (see Methods 3 for details).

Since the predictive uncertainties that the regressors provide are used actively in downstream applications (*e.g.* in the acquisition function of a Bayesian optimization procedure), we should assess our methods also on the quality of their uncertainty estimates. We can use the predictive uncertainties to assess *calibration* and *confidence* of a model [32–34]. The calibration implies that in expectation the average size of the prediction error should correspond to the magnitude of the grouped predictive uncertainties [35]. A method would thus be over-confident if the empirical error is larger than the uncertainties it predicts. We quantify this by computing a *standardized* (reduced) $\chi^2$ statistic: $\chi^2 = \frac{1}{N-1}\sum_i^N \left(\frac{y_i - \hat{y}_i}{\hat{\sigma}_i}\right)^2$ as the squared mean of the residuals normalized by the predictive variance, which provides us with one estimate for the regressor's calibratedness. We also investigate calibration curves [32, 33] by discretizing the predictions into $q$ quantiles of the predictive variances $\hat{\sigma}_i^2$ across all observations $i \in [1, n]$, indexed by $j \in [1, q]$, so that each $\sigma_j^2 \in \left[0, \frac{1}{q}, \ldots, 1\right]$ describes a Gaussian $\mathcal{N}_j(0, \hat{\sigma}_j^2)$. From this we compute *calibration*, *expected calibration error* (ECE), and sharpness (see Methods for definitions).

Plotting calibration assessments across regressors and representations (Fig 3) for our test systems (Figs G-J in S1 File), we find the methods generally to be well calibrated. Exceptions, *i.e.*

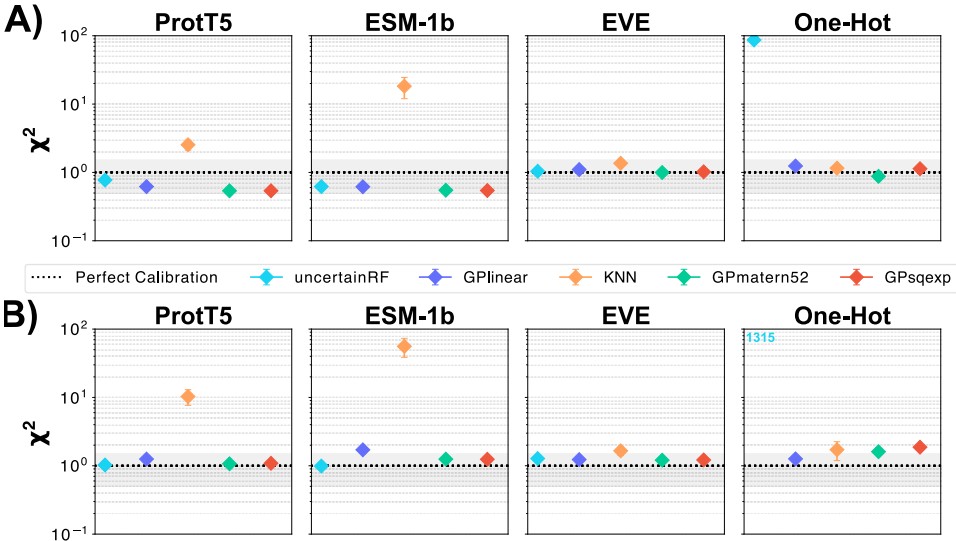

**Fig 3. Calibration of regressors.** Measured on $\beta$-LACTAMASE by the (reduced) $\chi^2$ statistic when training the regressors by splitting at random (A) compared to by position (B). Generally, the regressors are well-calibrated (grey region, dotted line is perfect calibration), slightly more so for the Position CV (B). Outliers are characterized by very low predictive variance values (see KNN regressor and the ONE-HOT representation).

overconfidence, occur if the regressor's predictive variance is very low, as is the case for kNN on PLMs or the RandomForest on ONE-HOT.

**Domains and generalization.** When we create a test set we implicitly define a notion of generalization: we assume that the test set is *as representative of the training set as we expect any future application to be*. The appropriate choice of test set will thus depend on the task we intend to solve with the model subsequently.

To make the discussion more precise, we will briefly formalize the concepts of *domain* and *task*. We define the *domain* as a finite set of embedded sequences $X \subset \mathbb{R}^d$ with the distribution $p(X)$. A test set is said to be out-of-domain if there is a shift in distribution between inputs in the training and test sets—sometimes referred to as *covariate shift*. A *task* involves a set of observations $Y \subset \mathbb{R}$, and is defined by the joint distribution $p(Y, X)$. The joint distribution factorizes as $p(Y, X) = p(Y|X)p(X)$ under the usual assumptions (for related transfer-learning definitions see [36–38]). With this definition, a task can change either through a covariate shift, or through a change in the likelihood $p(Y|X)$, *i.e.* a change in the fundamental relationship between $X$ and $Y$—sometimes referred to as *concept shift*. Note that even without a concept shift, a covariate shift will generally also lead to a shift in the output distribution $p(Y)$.

We fit a supervised model on a sample from a domain with the intent to make predictions related to a task, $f(\cdot) : \mathbb{R}^d \rightarrow \mathbb{R}$. The sample of size $n$ for the training of the supervised model is denoted $S|_{tr} = \{(X_i, y_i)\}_{i=1..n}$. To assess the performance on the task, we construct a test set as a representative subset of the task $\tilde{X} \subset \mathbb{R}^d, \tilde{y} \subset \mathbb{R}, S|_{te} = \{(\tilde{X}_j, \tilde{y}_j)\}_{j=1 n_{test}}$ with distributions $p(\tilde{X}), p(\tilde{Y}|\tilde{X})$, respectively. The generalization capabilities of the supervised model are assessed by residuals between the true observations and the predictions on the test data, as well as the calibration of predictive uncertainties.

The concept of domain is important in a protein engineering setting, because there is an inherent need for extrapolation. A typical engineering pipeline will gradually move further away from one or more initial (WT) sequences by introducing an increasing number of mutations. As an example of how these concepts can come into play, Fig 4 contains a schematic of

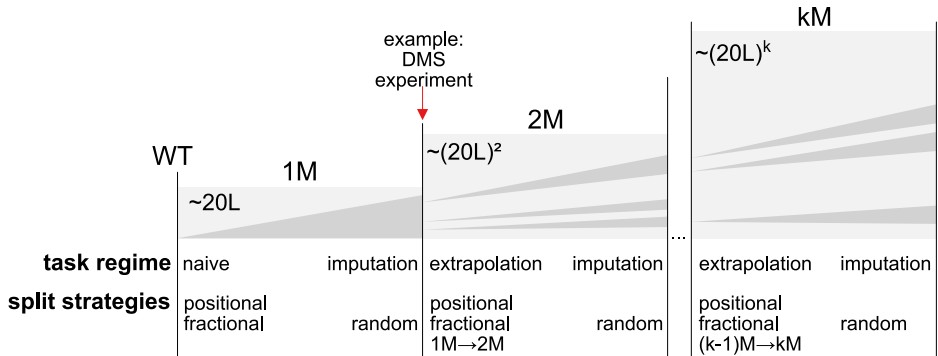

**Fig 4. Schematic data over domains of variants by number of mutations (1...*k*M) relative to sequence wildtype (WT).** Areas indicate observed variant samples (dark) against unexplored (light). Not all possible sequences can be explored for DMS experiments, due to natural constraints. The *task-regime* presents tasks applicable relative to the domain and the *split strategies* are potentially suitable protocols to assess performance on the domain by constructing training and test sets.

the available data in a typical protein engineering setting. The progression from left to right illustrates an exploration of variant space, organized by the number of mutations away from the wildtype (1M, 2M, . . ., *k*M). Within each block, the dark shade denotes an increasing percentage of cumulative variants explored. If we assume that the variants within each class are sampled according to some distribution, we can consider each block as representing a domain, for which we sample an increasing number of data points (left to right). The *DMS* arrow demonstrates the typical scenario explored in (Fig 2) where a Deep Mutational Scan (DMS) experiment provides us with a fairly complete set of single-variants. If we wish to predict missing values within the 1M class (and assume that these values are missing at random), then our task would be an imputation task. For this task, a random splitting strategy would be appropriate. In contrast, if we wish to make predictions in the 2M class, having only observed variants from the 1M class, we would be in an extrapolation setting, which is inherently a more difficult modelling task. A test set constructed on the 1M data using random splitting will not give a reliable estimate of our performance in this 2M prediction setting. The position-level splitter, which tests the ability of the model to make predictions at one position given only information at other positions, could serve as a more useful proxy for the performance on this task.

When observing experimental data for 2M and higher order variants, we often also face the problem that the data we encounter are biased. While the initial round of obtained variants is often sampled relatively uniformly (*e.g.* in a site saturation experiment), the subsequent double- and higher-order variants are often selected based on the best performing variants observed in earlier rounds (Fig 4, dark vs. light grey regions). Therefore, even if we define our domain to be all variants up to a maximum mutation order *k*, the sample bias in the data makes it difficult to reason about expected generalization. Another way of seeing this is that if we define our domain as a uniform distribution of variants up to order *k*, a site-saturation experiment of all 1M variants is a heavily biased sample.

**Assessing performance: Train/Test splits.** In our experiments comparing representations and regressors (Fig 2A and 2B), we saw a clear tendency towards lower performance when splitting by *positions* of a sequence compared to splits *at random*. We now discuss these two splitting strategies, in addition to several alternatives, in the light of the discussion above.

**Random CV.** This random splitting protocol is a standard *k*-fold cross-validation protocol that shuffles training data uniformly at random, separates ($X$, $y$) into $k$ folds, testing on

each after being trained on the remainder. When dealing with naturally occurring biological sequences, random splitting is often considered inappropriate due to the evolutionary structure in the data, and homology reduction schemes are therefore used for testing to better reflect how the model will generalize to sequences from species not included in the training set. For protein engineering where data consists of artificially constructed sequences, we do not have this concern, and there is therefore *a priori* no reason for disregarding random splitting. However, as argued above, it is appropriate only for imputing values that are missing at random, and will generally produce overly optimistic estimates of our performance beyond the imputation task.

**Positional CV.** This protocol partitions the sequence into segments of size $p$, such that we create $\frac{L}{p}$ number of splits. All variants with mutations that are in the range of positions in a segment are held-out for testing, while training is done on the remaining data. More specifically, let a sequence $X = \{x_0, \ldots, x_k, \ldots, x_L\}$ of length $L$ have a mutation at position $k$. The training set is then composed of the sequences that do not have a mutation at the given position range: $S|_{\text{tr}} = \{(X_i, y_i)\}_{i=1..N, k | k \notin [j, j+p]}$ (with partition starting index $j$) and $S|_{\text{te}}$ is the complement of that set. For computational feasibility, we consider positions in blocks of $p = 15$. This protocol therefore allows us to test generalization *beyond* the sequence positions observed during training. Rather than relying on a site-specific signal (the equivalent of conservation in natural sequences), the model must exploit pairwise or higher order signals. Since this assesses the ability of the model to predict *in the context* of the other amino acids present in the sequence, we might expect it to better reflect the performance we can expect in an extrapolation setting. On the other hand, if variant effects in a data set are largely additive, the position level splitter could provide an overly pessimistic view of our performance.

**Mutational CV.** This splitting strategy probes how well we generalize from a lower number of mutations to a higher number of mutations in the sequence. The protocol tests on a sample of degree-$k$ variants in the domain $kM$ containing $m$ variants, $S|_{\text{te}} = \{(X_j^{(kM)}, y_j^{(kM)})\}_{j=1..m}$, after having trained on variants of degree-$k'$ where $k' \leq k$, $S|_{\text{tr}} = \{(X_i^{(k'M)}, y_i^{(k'M)})\}_{i=1..n}$. In addition to an inherent shift in $p(X)$ there will often be an associated shift in $p(Y|X)$ when switching domains, as discussed above. The usefulness of the Mutational CV approach will thus depend on whether the variants predicted in expected application of the model are distributed similarly as the successive mutation degrees observed so far, *i.e.* if a similar selection protocol is used to select substitutions. Obviously, this splitting strategy is only feasible if the available data contains multiple substitutions, which is not the case for any of the data discussed so far. We will analyze such a case in detail below.

**Fractional CV.** We introduce this protocol to assess regressor performance as more data becomes available—from a naive, few observation setting, to the nearly full-information, imputation setting. The protocol is a $k$-fold CV protocol that sub-samples train and test data from the total available sequences uniformly at random. We sample each fraction $q \in (0, 1]$ uniformly at random $S|_{\text{tr}} \sim \{(X_i^{(j)}, y_i^{(j)})\}_{i=1..\lceil q \times N \rceil}^{j=1..k}$ with $k$ train, test iterations. This splitting strategy assesses the expected regressor performance when faced with different amounts of data within the domain. It thus serves as a simple proxy of the performance we can expect in a batch-1 optimization setting.

**Optimization.** Our final assessment strategy consists of an actual optimization protocol, where we use a Bayesian optimization strategy to actively select samples based on previous observations via the *Expected Improvement* [39] acquisition function. The protocol is similar to the Fractional-CV splitter, but replaces the simplifying assumption of uniform sampling of candidates (which is guaranteed to stay in-domain) with an active selection of candidates, thus becoming biased towards higher performance. Note that we in this protocol still optimize only

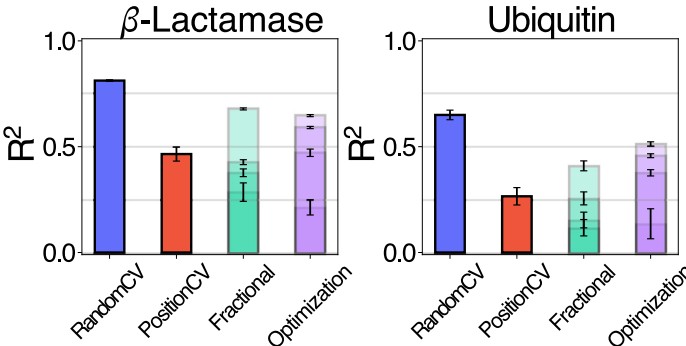

**Fig 5. Performance comparison across tasks.** GPMatérn$\frac{5}{2}$ regressor performance ($R^2$) for the ESM-1B representation of $\beta$-LACTAMASE and UBIQUITIN, when evaluated splitting at random (10-fold CV) (blue), by sequence positions (p = 15) (red), across fractions of the data (green), and optimization (purple). Fractional and optimization results are relative to available training data (four partitions each), with mean and std.err. reported respectively (from first partition (lowest value) to all available data (last partition, high value).

considering the known data points, with the task to find the optimal value in as few iterations as possible.

**Case 1: In-domain optimization.** We illustrate the Fractional-CV and Optimization splitter on the same datasets as before, comparing the performance in a single-mutation data setting, with increasing levels of data completeness (Fig 5); presenting the results as aggregates over quarters of the results (*i.e.* the performance across the first 25% of fractional splits and BO observations up to all available splits). We see that the Fractional-CV split and Optimization protocol give even more conservative assessments of the total performance (especially with only few available observations) than the earlier assessment criteria. Among the selection of regressors that we tested, the GPs again perform most reliably on this task, finding the optimal candidate after fewest iterations (Fig S in S1 File). This might be explained by the fact that the acquisition function of the optimization protocol relies actively on the uncertainty estimates provided by the regressors. We also note that random selection and sorting by EVE-score are competitive reference baselines (Figs S-T in S1 File).

In an ablation we investigate the parameters *k* and *p* for *random* and *positional* splitting respectively (Section 8 in S1 File) showing that the test performance metric is comparatively stable (Fig Q in S1 File). Note that in extreme cases (*e.g.* large *p*) resulting in fewer training data (and subsequently the larger the test set), the expected error and the standard error are likely to increase (Fig R in S1 File).

**Case 2: Extrapolation.** To illustrate some of the pitfalls that can occur when extrapolating to different domains, we conduct an analysis on a DMS experiment on PARD-ANTITOXIN [40]. This data set contains variants with multiple mutations, and is an example of the bias scenario introduced earlier, where covariate shifts lead to increasing shifts in the label distribution at higher mutation degrees.

In our analysis, we will use the *mutational CV* strategy introduced above, with the goal of estimating how well we can predict variants at increasing distance from the wildtype. Fig 6 demonstrates the performance obtained when training regressor models using data up to (or up to and including) the *k*th mutation degree, and testing on the *k*th. If we consider the mean square error (MSE) scores (Fig 6) (see Eq 4), the results behave as expected: when predicting on the 2M (or 3M) data, our performance increases if we include a randomly selected set of 2M (or 3M) variants in the training set as well. However, if we instead consider Spearman rank correlation (see Eq 5) as our metric, we see more surprising behavior: in the 3M case, our

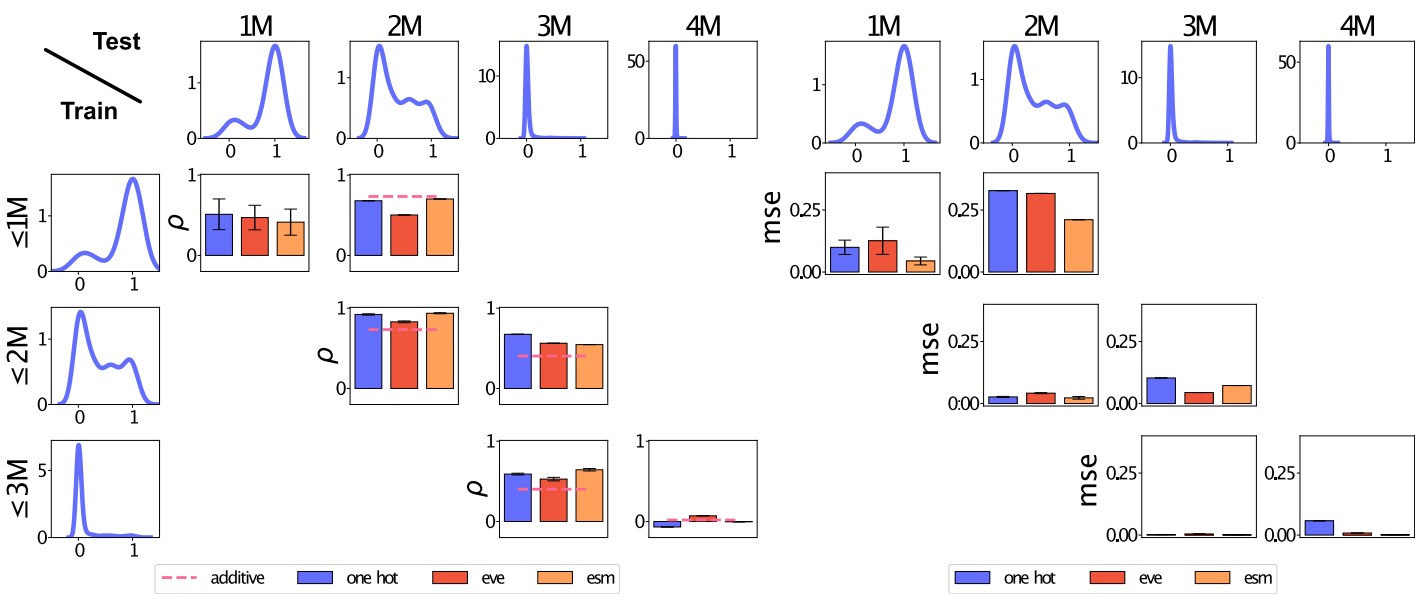

**Fig 6. *Mutation*-degree protocol on GP.** (Matérn$\frac{5}{2}$) from first-degree (1M) to fourth-degree (4M) of ParD-antitoxin represented by ESM-1B. Functional observations of ParD-antitoxin are density curves in first row and first columns respectively. We assess rank correlation (Spearman $\rho$) across in and out of domain (left test columns 1M to 4M). The reference is an additive benchmark baseline (dashed line), which is the addition for all variants constituting the target variants. The baseline for second-degree variants: y(Var1,Var2) = y(Var1)+y(Var2), and for triple variants the set of all combinations of the constituents. We assess accuracy (MSE) across in-domain and out of domain (right test columns 1M to 4M). The diagonal shows in-domain performance, such that we learn on sequences with less than or equal number of mutations, with a standard 5-fold CV protocol (given equal number of mutations randomly selected 20%). The off-diagonal for each source data domain (y-axis) shows what we predict in the next domain.

performance *decreases* if we include 3M samples in the training set. This inconsistent behavior is caused by the fact that the label distribution $p(Y)$ changes dramatically between the 1M, 2M, 3M and 4M settings. In the 1M case, we see a bimodal distribution between non-functional and functional with the latter being dominant. In 2M, 3M and 4M cases this switches more and more dramatically to the non-functional case, increasingly centering around a single value. As we move from 2M to 4M, the MSE therefore increasingly reports on the residuals around an almost constant prediction (in this dataset, none of the 4M mutations are functional). Since the empirical variance of the distributions drops, so do these residuals, giving a misleading impression of improved performance. Note that in this discussion we used MSE rather than $R^2$ to keep the discussion simple, since we for the $R^2$ would see shifts both in the numerator and the denominator (*i.e.*, the performance of the reference mean predictor also changes). See Fig M in S1 File for the equivalent $R^2$ plot.

In the Spearman correlation results we include as baseline the results from an additive model (dashed line), such that an observation of a double variant is predicted by the sum of the observations of its constituents. This simple procedure is often used in practice in engineering pipelines, in particular when optimizing for stability, where a sum of independent site contributions can be a reasonable approximation. We see that for this example, the additive model is in many cases a reasonable baseline for ranking candidates, although the additive values do not constitute competitive predictions in terms of MSE (Fig N in S1 File). Since ranking candidates is often a primary concern, this illustrates the importance of including such site-independent baselines when assessing regressor performance in the multi-variant setting.

**Case 3: Iterative data acquisition.** In a protein engineering campaign, experiments are often conducted iteratively. In this scenario, the selection of variants in a given iteration can

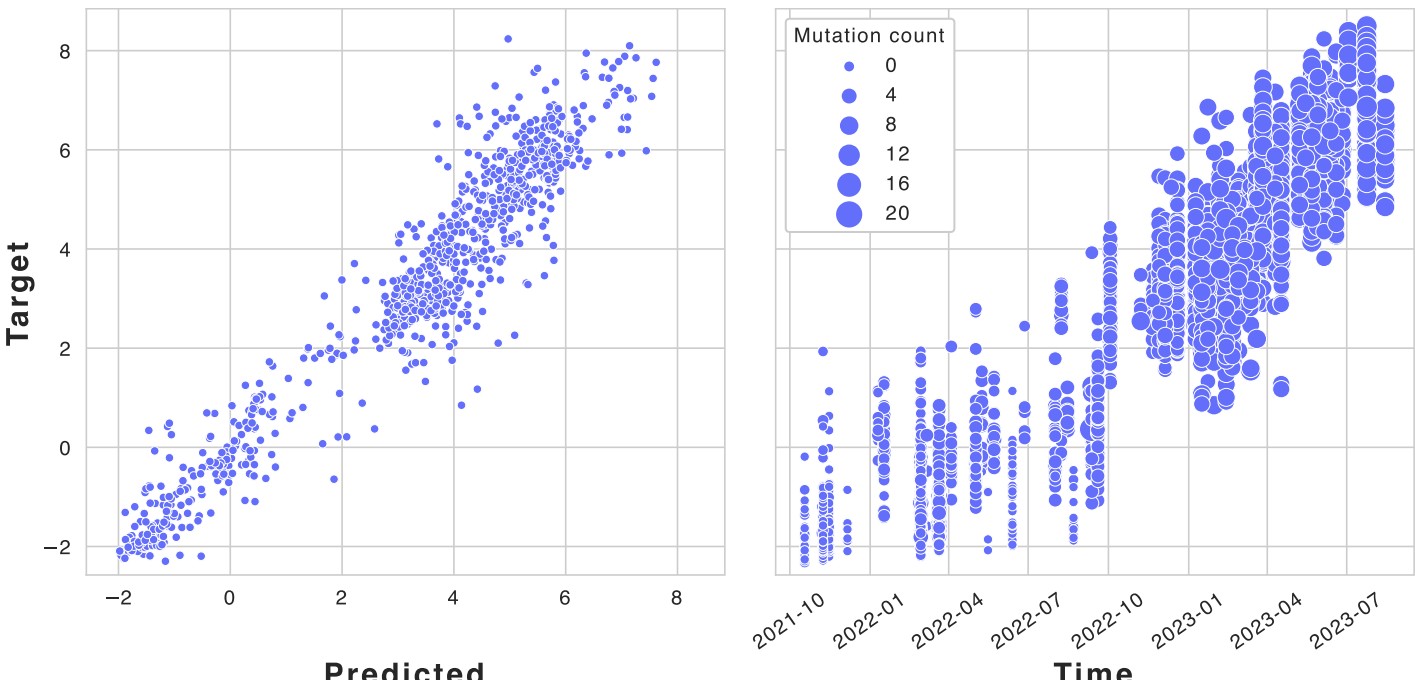

**Fig 7. The performance of a protein system with multiple mutations (0 to 21) collected iteratively.** The true values are plotted vs. the predicted values in a regression setting (left) and vs. time. It is critical to consider the number of mutations over time to properly model the system. The presented correlation in the regression setting is largely due to the selection bias from adding mutations over time, given the previous iterations.

depend on outcomes in previous iterations, over time leading to a gradual covariate shift. In the following, we show that this can have important consequences for the training and assessment of regression performance. We will illustrate this point on an in-house dataset. Although the identity of the protein system under study cannot be disclosed at this time, the case provides an interesting real-world example of the consequences of an ill-informed regression analysis. Fig 7 shows that a random split approach on the entire dataset gives rise to a correlation of 0.94 with experiment (using a Random Forest regressor with an ESM-2 embedding). A closer inspection reveals that the number of mutations in a variant is highly predictive of its fitness value: the further we are from the wildtype the higher fitness we obtain. Clearly, this is not a meaningful signal in the data: we do not expect higher order variants to generally have higher fitness value. The effect is caused by the iterative data acquisition strategy, where new variants are selected based on top-performing variants in the previous round—leading to improved variants over time. Due to a poor choice of splitting strategy, we have allowed our regressor to fit the selection bias in the data rather than the signal of interest. In this case, the proper splitting strategy is a chronological split in a typical forecaster scenario, where we train on the past and predict on the future. For this particular dataset, this decreased the obtained Spearman correlation from 0.94 to 0.19. Note that this is an extreme example of the sample bias issues discussed previously, where we see dramatic shifts in both $p(X)$ and $p(Y)$.

**Case 4: Free optimization.** So far, we have considered our test set as a perfect representation of a downstream task: our assessment measures how well we will do if the data in a downstream task is distributed identically to our test set. In reality, however, in a protein engineering pipeline, we conduct *free* (*i.e.* not bound to a predefined list of candidates) optimization where we *propose* candidates, and evaluate our regression algorithm on these proposals. It is therefore our responsibility to choose proposals such that they do not deviate too much

from the domain on which our regressor has been trained and validated. We can address this in various ways, either: 1) ensuring that the procedure used to select new candidates is restricted to points close to the training set, 2) assessing the proximity to the training domain by modelling the input distribution directly (*i.e.* with a density model $p(X)$), or 3) employing a regression model which will associate out-of-domain predictions with high degrees of uncertainty. Since the topic of our paper is regression, we will focus on the last point here, but note that examples of the first two options exist in recent protein optimization methods, where lists of candidates are generated to be close to wildtype proteins, for instance using a generative model of $p(X)$ [30, 31].

The question is thus if the uncertainties predicted by our regression algorithms are accurate enough to distinguish between useful and useless predictions. We saw that within the 1M domain, our different regression methods were all reasonably well calibrated (Fig 3), but since we only had access to 1M data, that analysis did not probe the out-of-distribution behavior. As a simple sanity check, we would expect that the uncertainties produced by our selection of regressors generally increases as we predict on sequences at larger distances from the wildtype. Unfortunately, we observe that this is only partially the case (Fig U in S1 File), as most of the methods produce constant uncertainty values at increasing distances. We had expected the GPs to shine in this area, but found that only the linear GP had a relative increase in predictive variance with respect to added mutations. Furthermore, the predicted variance values differ in orders of magnitude depending on the type of regressor and underlying representation. This observation is partially explained by a property of polynomial kernel functions (such as the linear kernel): the prior variance grows with the norm of the input [41 p. 90] which is not the case for stationary kernel functions (such as squared exponential and Matérn$\frac{5}{2}$).

The PARD-ANTITOXIN data allows us to investigate this effect in greater detail on real data. Here, we observe better calibration with more data (Fig P in S1 File), with generally a larger deviation from a perfect calibration compared to the previous assessment from data such as $\beta$-LACTAMASE, UBIQUITIN, *etc*. (Fig O in S1 File). However, the Gaussian Process models seem to most robustly quantify this effect for the multi-variant PARD-ANTITOXIN observations; such that they are better calibrated in this extrapolation setting.

An important consideration when discussing uncertainty quantification in protein optimization is that regression arises in two different contexts: 1) as surrogate models, for instance internally in a Bayesian optimization protocol, and 2) as oracles, which sometimes serve as the objective of optimization when experiments cannot be conducted, frequently used for method development of improved optimization schemes. Uncertainty plays different roles in these two settings. In the surrogate model setting, the uncertainty of predictions can reflect both epistemic (model) uncertainty and aleatoric (data) uncertainty. Our focus in Bayesian optimization is the epistemic uncertainty, which we wish to reduce by making additional observations. In contrast, in the oracle setting, the regressor is fixed during optimization, and serves as artificial, (*in silico*) experimental data. The uncertainty of a prediction made by the oracle should therefore be considered an aleatoric uncertainty, because it is irreducible once the oracle is trained. In the Bayesian optimization setting, we expect evaluations of the surrogate function regressor exactly in regions where the uncertainty is high (depending on the acquisition function), while in an oracle setting, it is appropriate to focus on regions where the regressor produces predictions with low uncertainty. A similar situation could arise in a real experimental setting if the experimental setup is known to provide very noisy outcomes in certain parts of the domain. In such cases, it would make sense in a Bayesian optimization protocol to incorporate the aleatoric uncertainty into the objective and acquisition function.

When using a regression algorithm as an oracle, an additional step of assessment is necessary to assess the candidates found during optimization, since we have no guarantees about the quality of an oracle evaluation. In particular, if the oracle was trained only on a limited size, system-specific dataset, we should be concerned that the optimization procedure has optimized for an extrapolation artefact of the oracle, rather than a signal from the data. In general, the free optimization setting implies that we have no data for the candidate sequence selected, so standard train/test splitting is not applicable. Ideally, experiments should be conducted at this stage to validate the candidates. However, in computational labs such experiments might not be accessible, which has led research towards finding computational proxies. One approach that has been suggested is to train two oracles on subsets of the data, optimize against one, and validate against the other—averaging over both permutations [31]. Ideally, the model assumptions should be different between the two oracles, to ensure that extrapolation artefacts are different between them. This approach to quantify the epistemic uncertainty of the oracle is similar to those built into the GP and RF regressors, but incorporates uncertainty about the model class as well, and could therefore provide more useful uncertainty quantification in the extrapolation setting.

## Discussion

In this study, we have identified the impact of some central components in a protein regression protocol: the data, the representations chosen to encode the data, the choice of model, and the assessment criteria. Our results confirm that some choices of representation can have positive impact on downstream regression performance, in particular those of large language models. The choice of regression algorithm itself seems to be less critical for performance, at least in the data-limited regime that we study here, and under the assumption that hyper-parameters of these models have been carefully tuned. This choice might therefore be guided by other considerations, such as whether the model can provide meaningful uncertainty estimates. In our experiments, Gaussian Process regression turned out to be particularly well suited for this task, although this is by no means the only option.

Of all the components in a regression protocol, assessment is perhaps the most critical. We demonstrate that performance can change dramatically depending on how data is split into train/validation/test sets. While this result is perhaps unsurprising, it warrants our attention, since performance values are routinely reported when new methods are published. If splits are not defined in the same way, comparing methods is meaningless. Equally important, if splits are defined too simplistically, we risk overestimating our abilities. A choice of splitting strategy implies an expected downstream task for which we are interested in quantifying expected performance. We give several examples of splitting strategies that attempt to reflect the regression scenario faced in a protein engineering protocol. The choice of assessment strategy is, however, complicated by selection bias in the data, and the inherent drive towards extrapolation when conducting protein engineering. We argue that for this reason, it is essential that we have the ability to associate predictions with reliable uncertainty estimates, and that we should evaluate regression methods on their ability to produce calibrated uncertainties.

Although we have stressed the importance of reliable uncertainty estimates in our analysis, the methods we considered were often lacking—especially when applied in an extrapolation setting. For Gaussian Process regression, this might be due to the high dimensionality of the inputs, and interesting future work could be to investigate whether lower dimensional approximations to learned embeddings might behave better in this setting. An alternative technique for uncertainty quantification worth exploring is *conformal prediction*. In particular, recent work in this area presents a solution to the bias that arises during optimization when iteratively adding proposals from a regressor to the available training data by an acquisition function [42].

Our previous discussion of domains and generalization indicate that it might not be fruitful to think about domains in terms of raw amino acid sequences and the number of mutations that a sequence is separated from the *WT*. If a method generalizes from 2M to 3M variants, it is presumably because it relies on *features* of the raw input that occur in both the 2M and 3M dataset, *e.g.* specific sequence patterns, charge distribution, or a combination of amino acids forming a stabilizing bond. Such features can easily be preserved between sequences deviating by tens of mutations. In a limited data regime, it can be difficult to reliably learn biochemically relevant features from the available data. Embeddings from *pre-trained* language models provide a potential shortcut for extracting such features, which explains why using them as input to a regression algorithm might improve performance. It is likely that these pre-learned representations also provide a richer notion of locality than that provided by raw input space. By making local perturbations in the representation of a protein, and decoding back to input space, one can hope to generate variants that are functionally or structurally close, but potentially more divergent in sequence space. This strategy has been employed by recent protein engineering pipelines [30]. A similar approach might prove useful in the future as a more robust basis for the definition of splits.

Given the large differences in performance arising from different split strategies and task definitions, we encourage the community to work towards standardized benchmarks that include splits of data that are well-motivated by specific biological tasks, including the extrapolative setting. Encouragingly, recent contributions seem to be moving in this direction. Apart from a standard cross-validation approach [19, 21, 23, 27, 43–45], previous work includes different hold-outs [11], homology splitting [13], mutational or blocks of positions splitting [24, 46, 47], and task-inspired splitting [24, 25, 47]. We stress that such benchmarks should include meaningful baselines (such as the additive model described above), and also rank models both by their ability to quantify prediction performance and uncertainty quantification. This will ensure that we can meaningfully compare performance of regression models, and provide practitioners with a realistic expectation of the performance of the state-of-the-art.

## Methods

### Computing representations

Seven distinct proteins were extracted from the ProteinGym dataset [14, 47].

The individual data are published in the following references: $\beta$-Lactamase [48], Ubiquitin [49], Calmodulin [50], TIM-Barrel [51], BRCA1 [52], T2-MTH [53], ParD-antitoxin [40].

The representations are obtained by different means: EVE is constructed from a single protein family multiple sequence alignment per protein (the UNLABELLED sequences in Table 1), for which we compute the latent space via the encoder of the EVE model [10] (see github.com/OATML-Markslab/EVE). We include the log-likelihood (ELBO) of the model, which has proven highly informative in an unsupervised setting (referred to as evo-score ($d = 1$)). To compute language models such as ProtBert, ProtT5, [26] and ESM-1b, ESM-1v, ESM-2 [11, 28] representations, we reduce the high dimensionality of the language model by taking the mean over the sequence positions from the last attention layer of the transformer model as our representation (as is frequently done in practice). We compare these learned representations with a simple ONE-HOT encoding of the raw input sequence.

### Optimizing regressors

We make the fairly common choice to standardize the observations in the training sample as $\bar{y} = \frac{y - \mu_y}{\text{std}(y)}$, where $\mu_y$ and $\text{std}(y)$ are the mean and standard deviation.

**Table 1. Overview of the datasets.**

| DATASET | REF. | UNLABELLED | LABELLED $N$ | LENGTH $L$ |
|---|---|---|---|---|
| $\beta$-Lactamase | [48] | 13191 | 4788 | 263 |
| Ubiquitin | [49] | 34281 | 1255 | 76 |
| Calmodulin | [50] | 37953 | 1729 | 149 |
| TIM-Barrel | [51] | 25264 | 1519 | 252 |
| BRCA1 | [52] | 9575 | 1184 | 237 |
| T2-MTH | [53] | 28232 | 1719 | 330 |
| ParD-antitoxin | [40] | 14672 | 9193 | 93 |

As regressors we fit:

1. a $k$NN regressor [54], as an example of a non-parametric, deterministic baseline, for which we optimize the number of neighbors [55],

2. a Random Forest (RF) as an example of a parametric, ensemble-based regressor [56], we implement the sklearn-implementation [55], which we extend with a predictive uncertainty estimate as $\hat{\sigma}^2 = \hat{y}^2 - \bar{\hat{y}}^2$,

3. a Gaussian Process (GP) regressor with: a linear kernel (equivalent to standard linear Bayesian regression [57]): $k_l(x, x') = \sum_{d=1}^{D} \sigma_d^2 x_d x_d'$,

4. a GP with a non-linear squared-exponential: $k_{SE}(x, x') = \sigma^2 \exp\left(-\frac{(x-x')^2}{2l^2}\right)$,

5. a GP with a Matérn-52 kernel: $k_{\frac{5}{2}}(x, x') = \sigma^2 \left(1 + \frac{\sqrt{5}(x-x')}{l} + \frac{5(x-x')^2}{3l^2}\right) \exp\left(-\frac{\sqrt{5}(x-x')}{l}\right)$.

We compute exact GP regression [41] (using GPFlow [58]) we use a zero-mean prior $f(x) \sim \mathcal{GP}(0, k(x, x'))$. We assume Gaussian noise for each experimental observation, such that

$$y_n = f(x_n) + \epsilon, \epsilon \sim \mathcal{N}(0, \sigma_\epsilon^2 \mathbf{I}). \tag{1}$$

The standardized observations do not affect RF or $k$NN predictions, and allows us to use a zero-mean prior for our GP models.

Given the GP setting, we can compute the marginal likelihood [41] for our training input $X$ of size $n$, as

$$p(\bar{y}|X, \theta) = -\frac{1}{2}\bar{y}^T [K_{XX} + \sigma_\epsilon^2 I]^{-1} \bar{y} - \frac{1}{2} \log |K_{XX} + \sigma_\epsilon^2 I| - \frac{n}{2} \log(2\pi). \tag{2}$$

The hyperparameters of the GP covariance functions were chosen as $\theta = \{\sigma^2, l, \sigma_\epsilon^2\}$ (the linear kernel parameters $\theta_l := \theta \setminus \{l\}$), such that $\sigma^2 \sim \Gamma^{-1}(3, 3), l \sim \Gamma^{-1}(3, 3), \sigma_\epsilon^2 \sim \mathcal{U}[0.01, 1.0]$ are loosely constrained. Given the standardized observation this is the least imposing prior that results in a more robust optimization, as it punishes unlikely extreme values for length-scale and variance. For each model the log-marginal likelihood is optimized using an L-BFGS optimizer (max = 500 iterations) [59].

We use skopt (scikit-optimize/stable) Bayesian optimization routine with an internal three-fold cross-validation, minimizing the negative mean absolute error (NMAE) to find the number of neighbors for the $k$NN regressor and number of estimators for the Random Forest [55]. The $k$NN regressor has $k \in \left[1, \lfloor 0.95 \frac{N}{3} \rfloor\right]$ with an optimization budget of 75 using gradient

boosted trees as internal surrogates (GBT). In the unoptimized setting we set number of neighbors $k = \lceil \frac{1}{3} N \rceil$. For the Random Forest regressor the number of estimators is $\in [2, N]$ with a budget of 15 (using GBT). The delta against Random Forest default parameters, and KNN fixed neighbors is in an ablation (see Fig V in S1 File).

## Protocols

The presented results for the RandomCV are obtained with a 10-fold cross-validation scheme (as implemented in [55]) unless specifically indicated otherwise. The PositionCV partitions the sequence data by positions. The fractional CV sub-samples the data available by the indicated fraction. For each fraction we compute a 5-fold CV, to obtain error estimates for each. More details with individual sizes of samples can be found in the Tables B and C in S1 File.

## Performance metrics and calibration

We compute the accuracy by an adjusted R2 score:

$$R2 = 1 - \frac{\frac{1}{N}\sum_i^N (y_i - \hat{y}_i)^2}{\frac{1}{N}\sum_i^N (y_i - \mu_y)^2}. \tag{3}$$

When not normalizing by training signal we resort to the elementary error metric as mean squared residuals:

$$MSE = \frac{1}{N}\sum_i^N (y_i - \hat{y}_i)^2. \tag{4}$$

To compute correlations we use the Spearman rank correlation (based on the ranks r of the inputs):

$$\rho = \frac{\mathrm{cov}(\mathrm{r}(y)\mathrm{r}(\hat{y}))}{\sigma_{\mathrm{r}(y)}\sigma_{\mathrm{r}(\hat{y})}}. \tag{5}$$

We compute calibration (goodness of fit) as a reduced $\chi^2$ statistic:

$$\chi^2 = \frac{1}{N-1}\sum_i^N (y_i - \hat{y}_i\hat{\sigma}_i)^2. \tag{6}$$

For details see SI Section 6. An implementation of the $\chi^2$ statistic can be found in the *evaluation* metrics in the *ProbNum* package [60]. As further uncertainty quantification assessments we compute confidence curves, as:

$$f_{cc}(\mathbf{Y}) = \sum_{j=1}^q \frac{1}{N}\sum_{i=1}^N l(\hat{y}_i, y_i)\mathbf{1}\{\hat{\sigma}_i \leq h_j\}. \tag{7}$$

The absolute deviation from calibration (ECE) and sharpness is:

$$\mathrm{ece}(f_{cal}) = \frac{1}{q}\sum_{j=1}^q |f_{cal} - j|, \tag{8}$$

$$\mathrm{sharp}(\hat{\sigma}^2_{1\ldots n}) = \frac{\mathrm{std}(\hat{\sigma}^2_{1\ldots n})}{\frac{1}{n}\sum_i^n \hat{\sigma}^2_i}.$$

(9)

## Bayesian optimization and the unsupervised baseline

We use the previously described regressors as surrogate functions and compute the expected improvement (EI) [61] as acquisition function, for the list of available labelled candidates from the dataset. The budget is 500 iterations, the regressor is optimized with the previously specified hyperparameters at each iteration. Ten different random seeds have been run for initial ordering of the available sequences. As reference baselines the EVE model evo-score was used to rank the available sequences (optimization results can be found in Fig S in S1 File).

## Supporting information

**S1 File. Supplementary material.** Contains dataset overview, additional results, calibration definitions, protocol descriptions.
(PDF)

## Author Contributions

**Conceptualization:** Richard Michael, Jacob Kæstel-Hansen, Simon Bartels, Jesper Salomon, Pengfei Tian, Nikos S. Hatzakis, Wouter Boomsma.

**Data curation:** Richard Michael, Jacob Kæstel-Hansen.

**Formal analysis:** Richard Michael, Jacob Kæstel-Hansen, Simon Bartels.

**Funding acquisition:** Richard Michael, Nikos S. Hatzakis, Wouter Boomsma.

**Investigation:** Richard Michael, Simon Bartels, Wouter Boomsma.

**Methodology:** Richard Michael, Jacob Kæstel-Hansen, Simon Bartels, Wouter Boomsma.

**Project administration:** Wouter Boomsma.

**Resources:** Nikos S. Hatzakis, Wouter Boomsma.

**Software:** Richard Michael, Jacob Kæstel-Hansen, Peter Mørch Groth, Simon Bartels.

**Supervision:** Jesper Salomon, Pengfei Tian, Nikos S. Hatzakis, Wouter Boomsma.

**Validation:** Richard Michael, Peter Mørch Groth, Simon Bartels, Jesper Salomon, Pengfei Tian, Wouter Boomsma.

**Visualization:** Richard Michael, Jacob Kæstel-Hansen, Peter Mørch Groth, Simon Bartels, Wouter Boomsma.

**Writing – original draft:** Richard Michael, Jacob Kæstel-Hansen, Wouter Boomsma.

**Writing – review & editing:** Richard Michael, Jacob Kæstel-Hansen, Peter Mørch Groth, Simon Bartels, Jesper Salomon, Pengfei Tian, Nikos S. Hatzakis, Wouter Boomsma.

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
