## [Decision Letter · Decision Letter 0]

25 Oct 2023

Dear Dr. Boomsma,

Thank you very much for submitting your manuscript "Assessing the performance of protein regression models" for consideration at PLOS Computational Biology. As with all papers reviewed by the journal, your manuscript was reviewed by members of the editorial board and by several independent reviewers. The reviewers appreciated the attention to an important topic. Based on the reviews, we are likely to accept this manuscript for publication, providing that you modify the manuscript according to the review recommendations.

Sincerely,

Piero Fariselli

Academic Editor

PLOS Computational Biology

Nir Ben-Tal

Section Editor

PLOS Computational Biology

Reviewer's Responses to Questions

**Comments to the Authors:**

Reviewer #1: in this study, the authors investigate different aspects of the application of machine-learning tools to optimization and engineering. Going through the different fundamental steps in a typical supervised machine-learning protocol, they explore challenges and issue related to data, encoding procedures, algorithm selection and performance evaluation. Overall the paper is well-written, the results are presented properly and conclusions drawn are interesting.

I only have the following minor concerns:

- I would extend the number of mutational datasets explored in this study, to provide a stronger and more general support to conclusions

- The choice of representative protein representation is in my opinion limited. Regarding evolutionary encoding, I would definitely include in the analysis sequence profiles/PSSMs as directly extracted from multiple-sequence alignments. For what concerns pLMs, the two selected models are in my view a little bit outdated: ESM1-v is probably more well-suited to mutation analysis than ESM1-b, while the most recent EMS2 is now available. Finally, ProtT5 would be more appropriate than ProtBert.

- Also the list of regression algorithms tested could be extended including SVR and/or deep multi-layer perceptrons

In summary, I believe that the paper could be generally improved just by providing a richer set of case studies, representation techniques and regression algorithms. I believe this would definitely add to the paper, providing a stronger support to conclusions.

Reviewer #2: The study investigates the impact of various components in protein regression protocols, underlining the significance of data representation and assessment criteria.

- correct typos. For instance, in Line 480, the statement "The individual data is published..." should be corrected to "The individual data are published...". Additionally, in Line 483, "The representations are obtained by different means.:" ends erroneously with both a period and a colon. The manuscript frequently employs parenthetical notations that disrupt sentence flow. Utilizing commas might enhance readability, for example: (approximate) in line 28, (variants) in line 62, and (sometimes referred to as wildtype exploration) in line 103.

- add reference for "e.g. the thermostability of an enzyme" mentioned in line 64.

- In Figure 2 A, while the bottom figure is clear, the upper portion's underlying concept lacks a clear elucidation in the manuscript. It's evident that various datasets and representation methods are being compared, but the meaning of the X and Y axes remains unclear.

- The exploration of distinct CV methods is intriguing. These CV methods are broadly outlined, yet the specifics of their fine-tuning throughout the manuscript remain nebulous. For instance, in Figure 5, is the randomCV reflective of a 10-fold or a 5-fold cross-validation? This detail should be explicitly mentioned when presenting results.

-An additional point related to the aspect above, in line with the paper's core theme: beyond illustrating discrepancies between CV methods, it would be insightful to demonstrate how certain parameters impact these features, like the effect of K in randomCV or the influence of p in positional CV.

-A discussion regarding the phenomenon of overfitting and how various choices can influence overfitting assessment would be a valuable addition to the manuscript.

**Have the authors made all data and (if applicable) computational code underlying the findings in their manuscript fully available?**

Reviewer #1: Yes

Reviewer #2: Yes

PLOS authors have the option to publish the peer review history of their article (what does this mean?). If published, this will include your full peer review and any attached files.

Reviewer #1: No

Reviewer #2: No

Figure Files:

Data Requirements:

Reproducibility:

References:

---

## [Decision Letter · Decision Letter 1]

10 Apr 2024

Dear Dr. Boomsma,

We are pleased to inform you that your manuscript 'A systematic analysis of regression models for protein engineering' has been provisionally accepted for publication in PLOS Computational Biology.

Best regards,

Piero Fariselli

Academic Editor

PLOS Computational Biology

Nir Ben-Tal

Section Editor

PLOS Computational Biology

Reviewer's Responses to Questions

**Comments to the Authors:**

Reviewer #1: authors have addressed all issues I raised

**Have the authors made all data and (if applicable) computational code underlying the findings in their manuscript fully available?**

Reviewer #1: Yes

PLOS authors have the option to publish the peer review history of their article (what does this mean?). If published, this will include your full peer review and any attached files.

Reviewer #1: No

---

## [Editor Report · Acceptance letter]

29 Apr 2024

PCOMPBIOL-D-23-01537R1 

A systematic analysis of regression models for protein engineering

Dear Dr Boomsma,

I am pleased to inform you that your manuscript has been formally accepted for publication in PLOS Computational Biology. Your manuscript is now with our production department and you will be notified of the publication date in due course.

With kind regards,

Anita Estes
